# Effects of urban green spaces on human perceived health improvements: Provision of green spaces is not enough but how people use them matters

Kowiyou Yessoufou[1]*, Mercy Sithole[1], Hosam O. Elansary[1,2,3]

1 Department of Geography, Environmental Management and Energy Studies, University of Johannesburg, Johannesburg, South Africa, 2 Plant Production Department, College of Food and Agriculture Sciences, King Saud University, Riyadh, Saudi Arabia, 3 Floriculture, Ornamental Horticulture, and Garden Design Department, Faculty of Agriculture (El-Shatby), Alexandria University, Alexandria, Egypt

* kowiyouy@uj.ac.za

## Abstract

How could we explain the mechanism driving the effects of Urban Green Space (UGS) on human health? This mechanism is a complex one suggesting, on one hand, an indirect effect of UGS Provision (measured as quantity, quality or accessibility of UGS) on health through UGS Exposure (measured as visit frequency to UGS, duration of visit or intensity of activities taking place during the visit). On the other hand, UGS Provision may have an indirect effect on Exposure, mediated by people's perception of UGS. The mechanism further suggests that UGS Exposure may influence indirectly human Health but mediated by human motivation to use UGS. We tested these different expectations by fitting 12 alternative structural equation models (SEMs) corresponding to four different scenarios, depending on how UGS Provision was approximated. We show that SEMs where i) Provision is approximated as UGS quantity, and Exposure as duration ($SEM_i$), ii) Provision is approximated as quantity, and Exposure as intensity ($SEM_{ii}$) and iii) Provision is approximated as distance of the closest UGS from people's house, and Exposure as intensity ($SEM_{iii}$) are equally the best of all 12 SEMs tested. However, apart from the $SEM_i$ that has no significant path, $SEM_{ii}$ and $SEM_{iii}$ have the same significant path (motivation ~ intensity; β = 7.86±2.03, p = 0.0002), suggesting that visits to UGS may be motivated by opportunities of physical activities offered by UGS. In all our scenarios, the best SEM is always the one where Exposure is measured as intensity, irrespective of how Provision is approximated. This suggests that it is not only UGS provision that matters the most in the mechanism linking UGS to human health improvement, but rather intensity, i.e. the type of activities people engage in when they visit UGSs. Overall, our findings support the theoretical model tested in this study.

**Data Availability Statement:** All relevant data are within the paper and its Supporting Information files.

**Funding:** This research was funded by King Saud University, Research Support Project, grant number RSP-2020/118 to HOE and National Research Foundation South Africa, grant number 112113 to KY. The funders had no role in study design, data collection and analysis, decision to publish, or preparation of the manuscript.

**Competing interests:** The authors have declared that no competing interests exist.

## 1. Introduction

Urban green spaces (UGSs) are open spaces in the public or private domains referring to all forms of greenery (parks, green roofs, woodlands, community gardens, lawns, sporting fields, bushes, ornamental plant arrangements, etc.) that are widely recognized as important in creating liveable cities [1–3]. They create an urban ecological systems [4] which contribute tremendously to the creation of sustainable cities. Green roofing, for example, is recently showed not only to mitigate urban temperature increase but also to maintain a climatically cool environment [5]. Similarly, other UGS types have been showed to regulate the regional thermal environment, e.g. in Dalian, China [6]. More critically, the lost of close to 60% of UGS area cover has been linked to the loss of close to 32% of the monetary values of the ecosystem services that these UGS provide to urban population in Ganjingzi, China [7]. Specifically on human health, UGSs have been reported to impact positively human health through a complex mechanism [8–11] illustrated in Fig 1 [10].

According to ref. [10], the effects of UGSs on human health are contingent upon human exposure to UGSs (referred to as "UGS exposure"), which is also a function of how many UGS is available in a given area (UGS provision). On one hand, UGS exposure can be measured in three ways, either as frequency (i.e. how often does one visit a UGS), duration (i.e. how long lasts a visit to a UGS) or intensity (i.e. what activity, passive or active, is undertaken during a visit). On the other hand, UGS provision can also be measured in three ways, including quantity (i.e. how many UGSs are there in a geographic area), quality (i.e., is a UGS qualitatively attractive?) or accessibility (i.e., is the access to UGS free or not or is it geographically close or far away from where people live). It is also important to highlight that the paths linking both UGS provision and exposure to health responses of human bodies are all influenced by a number of mediators (i.e., factors that promote UGS exposure or health response of human body) and moderators (i.e., factors that alter the strength of the relationships between different variables in the model; [10]). Zhang et al. [10] suggested that socio-demographic factors may play the role of moderator, e.g. education level. In the case of education, this is because education

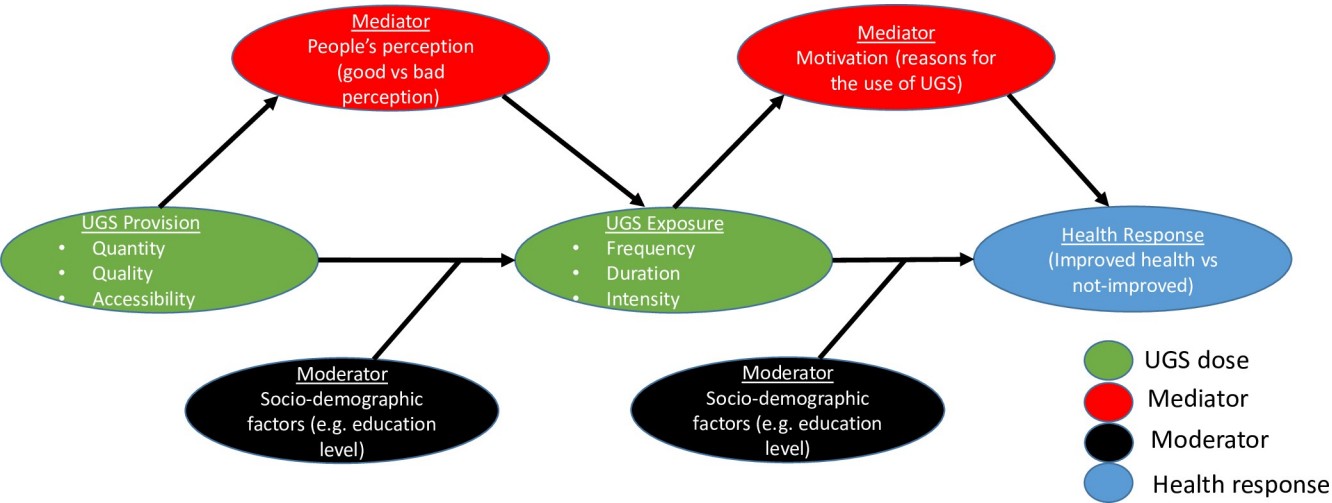

**Fig 1. Theoretical meta-model explaining the mechanisms driving the effects of Urban Green Space (UGS) on human health.** Each arrow symbolises a cause-effect relationship between two variables. On one hand, Provision of UGS (measured as quantity, quality and accessibility of UGS) has an indirect effect on human Health Response but through human Exposure to UGS (measured as visit frequency, duration of visit and intensity of activities taking place during a visit). On the other hand, Provision may have an indirect effect on Exposure mediated by people's perception of UGS, and UGS Exposure may have an indirect effect on human Health mediated by human motivation to use UGS. Adapted from ref. [10].

level may be determinant of people's level of understanding of UGS values, and this understanding would determine how consistent they are in using or not UGS. Given that provision and exposure can each be measured by three alternative metrics, 12 combinations of these metrics in four different scenarios are possible as illustrated in Fig 2.

Although the framework presented in Fig 1 [10] elucidates the theoretical paths linking UGSs to human health, there are few critical points that need to be highlighted and considered if we are to adopt the framework as universally applicable. Firstly, the framework is formulated based on existing quantitative literature on the topic but this literature is overwhelmingly exclusive to temperate and developed countries. As such, it may not necessarily be fully applicable outside the temperate region and the context of developed countries. Secondly, it remains possible that the framework, as it is formulated, may have been influenced by the views of those who proposed it. For example, no path linking directly UGS provision to health response was included (Fig 1). Thirdly, and more critically for data collection, all quantitative studies consulted in the process of formulating the framework used experimental lab-based data collection, which may involve the use of expensive equipment. For example, the effects of UGS on health were generally quantified using field experiments [12, 13] and UGS quantity was generally measured either as coverage area [14, 15] or average normalized difference vegetation index (NDVI) [16, 17]. Although these approaches aimed to generate more accurate dataset, it is time-consuming and requires sometimes expensive equipment as well as an important manpower, all of which could not always be afforded in studies aiming to collect data at global or regional scales. Finally, a scientifically accurate estimate of available UGSs

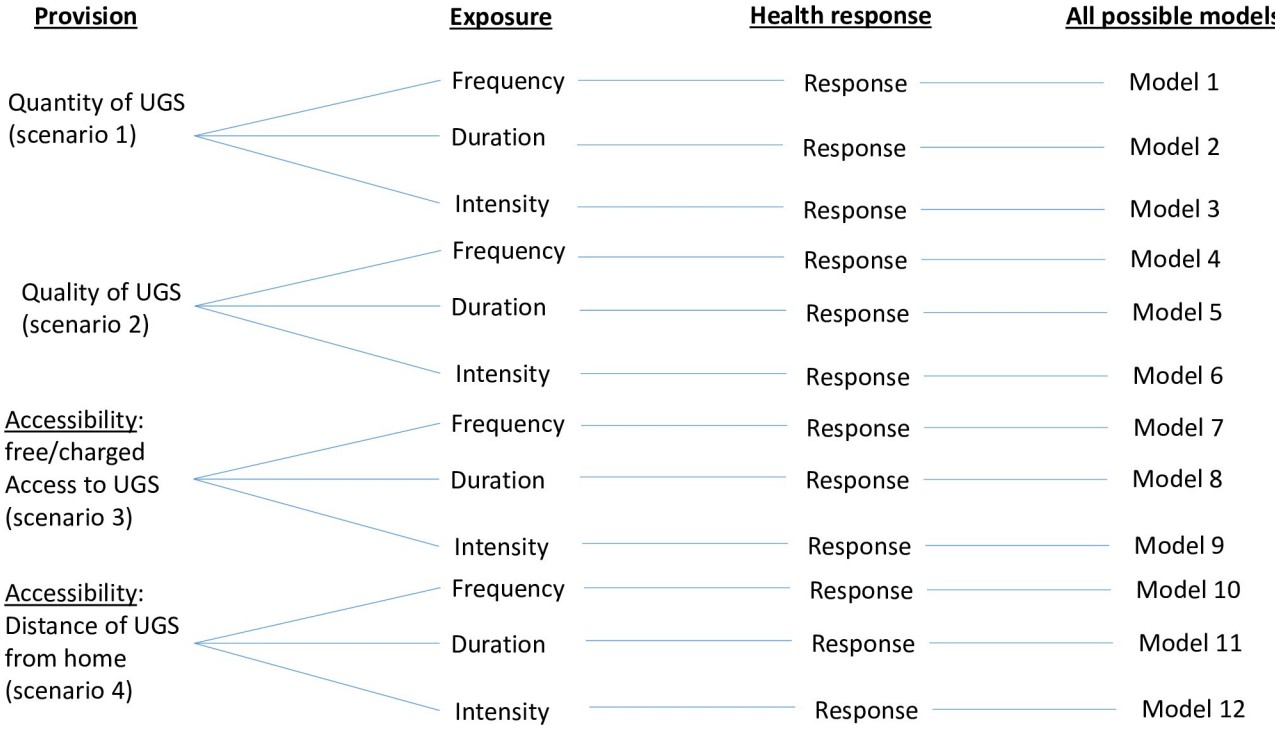

**Fig 2. All 12 possible models and four scenarios derived from all combinations of metrics of provision and exposure to UGS.** UGS provision can be measured as either UGS quantity (number of UGS in an area, Scenario 1), quality (Scenario 2), or accessibility (free/charged access, Scenario 3; distance from home, Scenario 4). UGS exposure can be measured as either frequency (how often does one visit an UGS), duration (how long does last a visit to an UGS) or intensity (active or passive activity during the visit). Health response is either Yes (UGS improves health) or No (UGS has no effect on health of our respondents). The combination of these variables leads to the definition of 12 models as illustrated in this Figure.

(UGS provision) does not mean that local people know about the existence of these UGSs in their areas and does also not mean that they actually do use them.

Consequently, we propose that measuring variables based on the perspectives of the UGS-users themselves is not only more meaningful but quicker and as such can lead to the collection of huge dataset even at large scale (global or regional). The rationale of our proposition is that people's use of UGS depends on their knowledge of UGS provision or availability (you cannot use something you do not know if it does exist and where it is and how many it is, etc.).

Therefore, the present study aims to use community-based dataset—a dataset based fully on people's own knowledge of UGS—to test the validity of one of the theoretical meta-models [10]. Specifically, the following questions were investigated. Given that 12 different variants of Zhang et al.'s meta-models are possible (Fig 2), are they all equally good? Which model is the best in each of the four scenarios possible (see Fig 2)? What are the significant paths in the overall best models and in the best model per scenario?

## 2. Material and methods

### 2.1. Ethics

This study was covered by the ethical approval from the Faculty Ethics Committee of the University of Johannesburg. The approval letter is submitted as supplemental Information.

### 2.2. Study area

The study site is the city of Bulawayo in Zimbabwe, southern Africa. Bulawayo is the second-largest city in Zimbabwe located in the Southwest part of the country. Located on a plain that marks the Highveld of Zimbabwe, the city is close to the watershed between the Zambezi and Limpopo drainage basins [18]. The population of Bulawayo is 1,200,750 with a female-to-male proportion of 52:48 [19]. The population in the province of Bulawayo is relatively young with 34% of the population being less than 15 years; only 3% are 65 years old and above [19]. Although several cultural groups are found in Bulawayo, the majority of residents belongs to the *Ndebele*) ethnic group followed by the *Shona* group [19]. A fraction of this population was interviewed during our data collection as explained below.

### 2.3. Data collection

**2.3.1. Selection of respondents and areas for data collection.**   Residential areas and local people were targeted for the collection of data pertaining to residents' perspectives on UGS-health relationships. To this end, sites selection was driven by their proximity to UGSs (e.g. parks) in different residential areas such as Cowdray Park, Luveve, Hillside, Mahatshula, Fortunesgate, Woodville, Nketa, Pelandaba, Bellevue, Selbourne Park, and the city centre. Then, a door-to-door visit to each of the households in these areas was conducted, and wherever people were available and willing to participate to the study, these people were selected for the interview. In addition, visits were conducted to the different UGSs in these areas, and people found in the UGSs were approached for interview. Lastly, random selection of people for interview was also done outside the above-mentioned residential areas. This random selection consisted of approaching anyone on the streets and asked him/her for interview after having explained the purpose of the study.

**2.3.2. Data collected.**   To assess residents' perspectives on UGS-health relationships, data on all variables included in Fig 1 were collected. This includes data on UGS Provision, UGS Exposure and health response.

Three metrics were used to approximate UGS provision: i) quantity of UGS, ii) quality of UGS and iii) accessibility of UGS. In the present study, quantity means how many UGS a respondent knows of, in the area where he/she lives. The quality of UGS was measured by the respondent's assessment of UGS quality on the following scale: zero, poor, average, good and high quality based on aesthetics, safety, and attractiveness of a specific UGS [20]. The accessibility of UGS was measured as either free/charged access to UGS or as an estimated distance (by the respondent) from a given UGS to the respondent's house.

To measure UGS Exposure, three types of data are also required: i) frequency of visit to UGS, ii) duration of the visit and iii) intensity of activities taken place in UGSs during the visit. Frequency implies how often the respondent visits an UGS; duration means how long lasts the visit on average and intensity implies whether the respondent conducted an active (e.g. sport) or not (passive activity; e.g. meditation) during the visit to UGS.

Data on health response following exposure to UGS were collected by asking the respondents the following question: Do you think that your health condition has improved since you have been visiting UGS? Health response is therefore a binary variable as the answer to the question is either YES or NO.

Finally, data on mediators and moderators were also measured. Two mediators were considered: i) people's perception of UGS (good or bad perception) and ii) motivation (reason for visiting or not a UGS). However, one moderator was taken into consideration, and this is the level of education of the respondents [10]. All data collected are provided in S1 Table.

**2.3.3. Mean of data collection and justification of the approach used.**   All data were collected exclusively through a semi-structured questionnaire (S1 File). Prior to any data collection, a potential respondent was first briefed about the project and then asked if he/she is willing to participate. If agrees, the respondent then signs an informed consent form. The study also meets all ethical requirements set by the ethical committee of the University of Johannesburg. Overall, 151 respondents participate freely to the present study. All data collected are presented in S1 Table.

The approach used in this study to collect information on all variables (Provision, Exposure and health response) differs from the approach used in most studies (see review in ref. [10]). The difference is that, in the present study, people's knowledge of UGSs and their own assessment of UGSs' impacts on health were given exclusive priority. This means that, instead of measuring, for example, UGS Provision using a classical metric (e.g. NDVI as a proxy for UGS quantity [16, 17], or measuring health response based on medical report, we rather measured these variables based on people's own knowledge of UGS provision as well as their own assessment of UGS influence on their health (health response). This approach has two advantages: first, it is very simple (no constraints of field experiment or lab-work and no need for expensive equipment), thus allowing the collection of data on all possible variables in Figs 1 and 2, and second, it prioritizes people's own perceptions/judgement of their own health condition instead of just medical report.

The rationale for prioritizing people's knowledge of UGS is that people's use of UGS depends primarily on their knowledge of UGS. These knowledge include the knowledge of not only UGS availability (you cannot use something you do not know if it exists or not, where it is and how many it is, etc.) but also human's personal judgement of UGS effects on health (no one would use a UGS if they do not see the need for it).

**2.3.4. Data analysis.**   All data were analysed in R 3.5 [21], and the R code used are presented in S2 File.

Prior to the analysis, some categorical variables are coded as numeric. For example, the respondents expressed frequency of visit to UGS as daily, weekly, monthly and occasionally. These visit frequencies were converted into 30, 4, 1 and 0.5, respectively: daily visit to UGS was approximated as 30 days visit a month; weekly means 4 times a month; monthly was assumed

to be once a month, and the value 0.5 was attributed to occasional visit. Quality of UGS was coded numerically as follows: zero (0), poor (1), average (2), good (3), and high quality (4). Health responses were coded as 0 (No, there is no improvement of health) and 1 (Yes, there is improvement). People's perceptions of UGS was coded as good (1) or bad (0).

Then, each of the relationships or paths in Fig 1 was translated into a *GLM* model (Generalized Linear Model, see R code in S2 File) and 12 structural equation models (SEMs) were fitted to all data collected using all GLM models in the ref.'s [10] meta-model. These SEMs, fitted using the R library *piecewiseSEM* [22] contain four *GLM* models with a binomial error structure when the response variables are binary (e.g. health response). The 12 SEMs fitted correspond to all possible alternative combinations of variables, given that each variable can be approximated with at least 2 metrics (e.g. Provision and Exposure are each measured with three metrics; Fig 2). The adequacy of each SEM is tested based on its Goodness of fit (C value) and P value. An adequate SEM to the data is expected to show the lowest C value possible and a $p > 0.05$ [22]. The best of all models was identified based on AIC value, and the significant paths in each model were identified when $p < 0.05$.

Finally the selection of the best models was based on AIC value with a threshold value of 3.

## 3. Results

The results of all 12 meta-models fitted to the data are presented in S2–S13 Tables and are summarized in Table 1. The first question explored was: Is Zhang et al.'s models a good fit for the data collected. Irrespective of how provision and exposure were measured, the analysis shows that any of the 12 meta-models can be used to explain the relationships between UGS and health condition (that is, for all 12 models, $P > 0.05$; see column "*Fitness of the model*" in Table 1 and also note in this Table that there is no significant missing path).

If all 12 models are a good fit to the data, the next question is: Are they all equally good? The results show that only meta-model 2 ($AIC_{model2} = 52.02$), meta-model 3 ($AIC_{model3} = 52.48$) and meta-model 12 ($AIC_{model12} = 53.49$) (see definitions of each model in Fig 2 and S1 Table) are equally the best of all 12 meta-models tested (Table 1). These three meta-models are the best because the differences between their respective AIC values and each of the AIC of the remaining nine meta-models are greater than 3, but they are equally good because the difference between these three models is $\Delta(model 2,3,12) < 3$ (Table 1).

An additional but important question is: Which model is the best in each of the four scenarios in Fig 2? Irrespective of the scenarios considered, the best model is always the model where Exposure is measured as intensity, although in some scenarios, the difference between AIC of the best model and the rest of the models is only marginal (see column "*Fitness of the model*" in Table 1).

Finally, what are the significant paths in the overall best models and in the best model per scenario? In the overall three best models, apart from the model 2 that has no significant path, models 3 and 12 have the same significant path (motivation ~ intensity; $\beta = 7.86 \pm 2.03$, $p = 0.0002$; Table 1). When looking at scenario per scenario, this significant path still remains the same except for scenario 2 where the significant path is motivation ~ quality ($\beta = -2.47 \pm 0.82$, $p = 0.003$; Table 1).

## 4. Discussion

### 4.1. Explaining patterns of people's judgement of their health responses to UGS

The analysis shows that any of the 12 models can be used to explain the relationships between UGS and health condition. This is a strong first evidence that validates the meta-model or

**Table 1. Summary of path coefficients of the 12 models tested in this study.** Each path corresponds to an arrow on Fig 1, and the definitions of the 12 models and four scenarios are illustrated in Fig 2.

| Provision | Exposure | Fitness of the model | | | | Significant paths | Coefficient significant path | Significant missing path |
|---|---|---|---|---|---|---|---|---|
| | | C value | DF | P value | AIC | | | |
| Quantity (Scenario 1) | Frequency (model 1) | 17.1 | 18 | 0.51 | 63.1 | Frequency ~ perception_in_relation_to_health | β = -7.62±3.08, P = 0.01 | none |
| | | | | | | health_response~ education_level:quantity | β = 1.26±0.54, P = 0.01 | |
| | | | | | | health_response~ quantity | β = -3.41±1.48, P = 0.02 | |
| | | | | | | health_response~ education_level | β = -1.80±0.83, P = 0.03 | |
| | Duration (model 2) | 12.02 | 22 | 0.95 | 52.02 | none | NA | |
| | Intensity (model 3) | 14.48 | 22 | 0.88 | 52.48 | mediator_motivation~ intensity | β = 7.86 ±2.03, P = 0.0002 | |
| Quality (Scenario 2) | Frequency (model 4) | 15.2 | 16 | 0.51 | 59.2 | frequency_in_a_month ~ quality | β = -4.10±1.09, P = 0.0003 | none |
| | | | | | | mediator_motivation ~ quality | β = -1.99±0.96, P = 0.04 | |
| | Duration (model 5) | 16.6 | 16 | 0.41 | 60.6 | duration_hour ~ quality | β = 0.62±0.09, p<0.001 | none |
| | Intensity (model 6) | 14.8 | 16 | 0.53 | 56.8 | mediator_motivation ~ intensity | β = 8.22±1.96 P = 0.0001 | none |
| | | | | | | mediator_motivation ~ quality | β = -2.47±0.82 P = 0.003 | none |
| Accessibility: | | | | | | | | |
| 1. Free/not (Scenario 3) | Frequency (model 7) | 21.22 | 18 | 0.26 | 73.22 | frequency_in_a_month ~ access_charged | β = -9.44±3.14 P = 0.0034 | |
| | | | | | | frequency_in_a_month~ perception_in_relation_to_health | β = -6.61±2.99 P = 0.029 | |
| | Duration (model 8) | 10.77 | 18 | 0.90 | 62.77 | duration_hour ~ access_charged | β = 1.77±2.56, P<0.001 | none |
| | Intensity (model 9) | 11.27 | 18 | 0.88 | 61.27 | mediator_motivation ~ intensity | β = 7.86±2.03, P = 0.0002 | none |
| 2. Distance (Scenario 4) | Frequency (model 10) | 14.94 | 18 | 0.66 | 60.94 | frequency_in_a_month ~ access_distance | β = -0.02±0.004, P<0.001 | none |
| | | | | | | frequency_in_a_month ~ perception_in_relation_to_health | β = -7.36±2.84, P = 0.01 | |
| | Duration (model 11) | 11.18 | 18 | 0.88 | 57.18 | duration_hour ~ access_distance | β = -0.001±0.0004, P = 0.004 | none |
| | Intensity (model 12) | 11.49 | 18 | 0.87 | 53.49 | mediator_motivation ~ intensity | β = 7.86±2.03, P = 0.0002 | none |

framework proposed by ref. [10] to explain the overall mechanism through which UGSs may influence human health. This also implies that data collected based on people's knowledge and perceptions of the effects of UGS on health may be used to accelerate massive data collection in the future. However, not all 12 models are equally good; models 2, 3 and 12 outperform all other models. These three best models correspond to different scenarios. On one hand, they correspond to the scenario where UGS provision is approximated as quantity, and exposure as either duration of visit to UGS (model 2) or intensity (active or passive activity taken place during the visit to UGS) (model 3). This is perhaps indicative of the importance of how people who have knowledge of the number of existing UGSs (quantity) use these UGSs in term of

duration of their visit and the type of activity (active or passive) they engage in during the visit. On the other hand, they correspond to a scenario where UGS provision is measured as distance (distance from home to UGS), and exposure as again intensity (model 12). This scenario reveals perhaps how important is the distance effect in determining the type of activities taken place during the visit to UGS [10].

Furthermore, the investigation of each scenario reveals that the best model in each scenario is always the model where Exposure is measured as intensity, irrespective of how provision is measured. This is a key finding as it suggests that it is not UGS provision only (number of UGS, its quality or accessibility) that matters most, but rather intensity, i.e. the type of activities people engage in when they visit UGSs. As showed consistently in all best models, intensity is itself predicted positively and significantly by motivation, a mediator variable defining the reason for the use of UGS [10]. This positive relationship implies that people who are motivated to visit UGS are more likely to engage in active exercise (e.g. sport, gym, walking, etc.) during their visit than people who do not have any specific motivation (e.g. people who are simply invited by a friend to a UGS or people who are just passing by). Interestingly, there was too a positive significant relationship between quality of UGS and intensity in scenario 2, suggesting that the type of activities taken place during a visit to UGS is not only determined by the motivation of the visit to UGS but also by the quality of UGS. This makes sense as the definition of quality of UGS includes the presence and quality of e.g. sport infrastructure [20].

Indeed, several studies have reported that the quality of green spaces not only influences the likelihood of physical activity taking place during a visit to UGS but also influences the frequency of these activities [8, 23–25]. These previous findings are supports for the findings reported in the present study. Earlier studies showed that people living close to UGSs are more likely to visit them for physical activity [26]. This shows the importance of distance in the use of UGS as revealed in model 12, one of the three best models in the present study. Several other studies have made such link between distance to UGSs, the likelihood of residents using them for physical activities and the lower incidence of obesity and heart diseases [8, 26]. However, ref. [24] disputed the notion of association between physical activity and access to UGS and argued that, although UGSs do have positive associations with the perceived general health of residents, availability of UGS is not a determinant of total physical activity [24]. The present study is in support to ref. [24] as it reveals that, not only the distance is key but the UGS quality is too important. UGS quality, here, refers to UGS that provides walking or cycling trails and venues to play and exercise, and such quality UGS was reported as motivating people to use them for the benefit of their health conditions [20, 27–29].

## 4.2. Differences in approach used in the present study versus previous studies

The approach used to measure different variables in ref.'s [10] model is different from approaches used in previous studies. For example, in several studies, health responses were assessed in various ways. These include self-reported health, the 12-item General Health Questionnaire [30], Kessler 6 instrument [31], Mental Health Inventory [32], and the 36-Item Short Form Health Survey [33]. They also include health-related complaints (headaches, nausea, dizziness, listlessness etc.) in the last 14 days [32, 34], and visits to mental health specialists and intake of medication [35] as well as obesity. Also, the effects of UGS on health were quantified using either field experiments [12, 13, 36–38] or longitudinal data sets [30, 39–42]. However, in the present study, a simplest approach was used, that is, people's own feeling or assessment of change in their own health condition since they have been visiting UGS is used as proxy for health response to UGS. This proxy is what others refer to as perceived general condition [24, 34].

Also, the present study differs in the way the metrics of UGS provision was assessed. While people's own assessment of the UGS provision was prioritized in this study, other studies developed some complex approaches. For example, the UGS quantity was generally measured in two ways: coverage area [14, 15] or average normalized difference vegetation index (NDVI) [16, 17] whereas in the present study, UGS quantity was measured as the number of UGS known by the respondents in the area where they live. Also, while the present study assessed UGS accessibility as whether access to UGS is free or charged or as the estimated distance of UGS location from the respondents' house, this latter approach is also used in the literature. For example, ref. [15] used the average distance by road network from all addresses in the neighbourhood to the nearest green spaces whereas others have used the travel time by car to the nearest green space [42].

As far as the UGS quality is concerned, several approaches are also adopted in different studies. Some assessed UGS quality as a weighted mean score of ten attributes (facilities, shade, water features etc.) [31]; others used audit tools [31, 32, 43], or percentage of respondents who consider the quality of the green spaces as good [15]. The approach used in the present study is similar to the ref. [15]'s approach with the particularity that the respondents provide their own assessments of the UGS quality on a 5-scale rank (zero quality, poor, average, good, and high quality). Despite these differences, the fact that ref. [10]'s framework fits well to the data collected in the context of Bulawayo means that our simple approach to data collection may be used for further investigations of the UGS-health relationships.

### 4.3. Conclusion

Understanding the mechanisms linking improvement of human health conditions to UGS is key to promote UGS for healthy cities and their use by human. To this end, ref. [10] proposed, based on quantitative data collected through various means in temperate and developed world, a theoretical framework and various metrics with which they explained these mechanisms. We suggest that, in tropical and poor countries where lab. equipment and financial support for research are limited but the need for healthy cities is growing, using a questionnaire-based data collection can be easily used to investigate UGS-health relationships and promote the establishment of UGS in urban areas. Nonetheless, we acknowledge that, due to the way health response was measured in the present study, our health response may not necessarily correspond always to the diagnostic on a medical report, hence we referred to it as perceived health conditions by the respondents.

## Supporting information

**S1 File. Questionnaire used for data collection.**
(DOC)

**S2 File. R script used for data analysis.**
(DOC)

**S1 Table. All data collected and analyzed in this study.**
(DOC)

**S2 Table. Path coefficients of meta-model 1 defined in Fig 2.** See R scripts in S2 File for details of the meta-model. * indicates significant relationships between predictor and response.
(DOC)

**S3 Table. Path coefficients of meta-model 2 defined in Fig 2.** See R scripts in S2 File for details of the meta-model. * indicates significant relationships between predictor and response.
(DOC)

**S4 Table. Path coefficients of meta-model 3 defined in Fig 2.** See R scripts in S2 File for details of the meta-model. * indicates significant relationships between predictor and response.
(DOC)

**S5 Table. Path coefficients of meta-model 4 defined in Fig 2.** See R scripts in S2 File for details of the meta-model. * indicates significant relationships between predictor and response.
(DOC)

**S6 Table. Path coefficients of meta-model 5 defined in Fig 2.** See R scripts in S2 File for details of the meta-model. * indicates significant relationships between predictor and response.
(DOC)

**S7 Table. Path coefficients of meta-model 6 defined in Fig 2.** See R scripts in S2 File for details of the meta-model. * indicates significant relationships between predictor and response.
(DOC)

**S8 Table. Path coefficients of meta-model 7 defined in Fig 2.** See R scripts in SI-4 for details of the meta-model. * indicates significant relationships between predictor and response.
(DOC)

**S9 Table. Path coefficients of meta-model 8 defined in Fig 2.** See R scripts in S2 File for details of the meta-model. * indicates significant relationships between predictor and response.
(DOC)

**S10 Table. Path coefficients of meta-model 9 defined in Fig 2.** See R scripts in S2 File for details of the meta-model. * indicates significant relationships between predictor and response.
(DOC)

**S11 Table. Path coefficients of meta-model 10 defined in Fig 2.** See R scripts in S2 File for details of the meta-model. * indicates significant relationships between predictor and response.
(DOC)

**S12 Table. Path coefficients of meta-model 11 defined in Fig 2.** See R scripts in S2 File for details of the meta-model. * indicates significant relationships between predictor and response.
(DOC)

**S13 Table. Path coefficients of meta-model 12 defined in Fig 2.** See R scripts in S2 File for details of the meta-model. * indicates significant relationships between predictor and response.
(DOC)

## Acknowledgments

The authors extent their appreciation to King Saud University for assistance (RSP-2020/118) as well as all the respondents to our questions during data collection.

## Author Contributions

**Conceptualization:** Kowiyou Yessoufou.

**Data curation:** Mercy Sithole.

**Formal analysis:** Kowiyou Yessoufou.

**Funding acquisition:** Kowiyou Yessoufou, Hosam O. Elansary.

**Investigation:** Kowiyou Yessoufou, Mercy Sithole, Hosam O. Elansary.

**Methodology:** Kowiyou Yessoufou.

**Project administration:** Kowiyou Yessoufou.

**Resources:** Kowiyou Yessoufou.

**Supervision:** Kowiyou Yessoufou.

**Validation:** Kowiyou Yessoufou, Hosam O. Elansary.

**Writing – original draft:** Kowiyou Yessoufou.

**Writing – review & editing:** Hosam O. Elansary.

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
