## [Decision Letter · Decision Letter 0]

10 Jul 2020

PONE-D-20-17738

Testing a model explaining the mechanism driving the effects of urban green spaces on human health

PLOS ONE

Dear Dr. Johannesburg,

Thank you for submitting your manuscript to PLOS ONE. After careful consideration, we feel that it has merit but does not fully meet PLOS ONE’s publication criteria as it currently stands. Therefore, we invite you to submit a revised version of the manuscript that addresses the points raised during the review process.

We look forward to receiving your revised manuscript.

Kind regards,

Jun Yang

Academic Editor

PLOS ONE

Additional Editor Comments:

Reviewer 1

Overall, this paper is well-written and the topic is interesting. However, I suggest authors do some necessary revisions.

(1) Overall, the English is good, while some expression is informal. On one hand, on the other....

(2) The Title of this paper should be better if you use 'The mechanism driving the effects of urban green spaces

on human health: based on structural equation models'

(3) The abstract is actually not well written. line 12-19, authors focus on the model results, while it requires your efforts to translate into the general findings, as some readers are not familiar with the models.

(4) Line 30-35, the description is too short, you may try to describe in detail, especially the human health effect (e.g. noise reduction, urban heat island mitigation, urban flooding management, air and water purification). This is because your paper is focusing on the health effect.

You may refer:

1. The maintenance of prefabricated green roofs for preserving cooling performance: A field measurement in the subtropical city of Hangzhou, China. Sustainable Cities and Society, 61, 102314.

2. Assessing the Impacts of Urbanization-Associated Green Space on Urban Land Surface Temperature: A Case Study of Dalian, China[J]. Urban Forestry & Urban Greening. 2017(22), 1–10.doi:10.1016/j.ufug.2017.01.002.

3.Spatiotemporal variations in greenspace ecosystem service value at urban fringes: A case study on Ganjingzi District in Dalian, China[J]. Science of the Total Environment. 639 (2018) 1453–1461,.doi:10.1016/j.scitotenv.2018.05.253.

(5) line 95-98, not needed, as you mention it in the end of this paper.

(6) for statistic analysis, the p-value, should not be capitalize

Reviewer 2

This paper explains the driving mechanism of the effects of urban green spaces on human health through structural equation models (SEMs). It has interesting findings such as that intensity (types of activities people engage in urban green spaces) matters the most to human health improvement. I have some comments and suggestions for the authors to consider to further improve the paper.

First, while this is an interesting paper examines an important question, I am not clear about the contribution the authors made to the field. The authors mentioned that they use Zhang et al’s model as the theoretical backbone. If so, please explain Zhang et al’s model in detail and point out what improvement the authors have made. Currently, it is not clearly if the authors’ model is any different from Zhange et al.

Second, the authors shall provide justification to their models and data variables. The biggest difference of this paper’s data variables, as the authors mentioned, is that they use people’s knowledge of UGS and effects on health, rather than other classical metrics. While this has certain advantages, such as simple and easy to measure, it also has subjective bias. The authors shall discuss this limitation. It also needs to provide justification why they prioritize people’s own perceptions/judgement over more objective measures. On another issue, the authors used education as a moderator, but I cannot find any reference or any justification in the text why education shall be considered as a moderator.

Finally, health response is measured subjectively and shall be renamed as “perceived health improvement” or other similar name.

Minor issues:

female-to-male proportion of 52% and 48%, it shall be 52:48 (2.2 study area)

Journal Requirements:

2. Please amend the manuscript submission data (via Edit Submission) to include author Kowiyou Yessoufou.

3. Your ethics statement must appear in the Methods section of your manuscript. If your ethics statement is written in any section besides the Methods, please move it to the Methods section and delete it from any other section. Please also ensure that your ethics statement is included in your manuscript, as the ethics section of your online submission will not be published alongside your manuscript.

4. Please include your tables as part of your main manuscript and remove the individual files. Please note that supplementary tables (should remain/ be uploaded) as separate "supporting information" files.

Reviewers' comments:

Reviewer's Responses to Questions

**Comments to the Author**

1. Is the manuscript technically sound, and do the data support the conclusions?

Reviewer #1: Yes

Reviewer #2: Partly

2. Has the statistical analysis been performed appropriately and rigorously? 

Reviewer #1: Yes

Reviewer #2: Yes

3. Have the authors made all data underlying the findings in their manuscript fully available?

Reviewer #1: Yes

Reviewer #2: Yes

4. Is the manuscript presented in an intelligible fashion and written in standard English?

Reviewer #1: Yes

Reviewer #2: Yes

5. Review Comments to the Author

Reviewer #1: Overall, this paper is well-written and the topic is interesting. However, I suggest authors do some necessary revisions.

(1) Overall, the English is good, while some expression is informal. On one hand, on the other....

(2) The Title of this paper should be better if you use 'The mechanism driving the effects of urban green spaces

on human health: based on structural equation models'

(3) The abstract is actually not well written. line 12-19, authors focus on the model results, while it requires your efforts to translate into the general findings, as some readers are not familiar with the models.

(4) Line 30-35, the description is too short, you may try to describe in detail, especially the human health effect (e.g. noise reduction, urban heat island mitigation, urban flooding management, air and water purification). This is because your paper is focusing on the health effect.

You may refer:

1. The maintenance of prefabricated green roofs for preserving cooling performance: A field measurement in the subtropical city of Hangzhou, China. Sustainable Cities and Society, 61, 102314.

2. Assessing the Impacts of Urbanization-Associated Green Space on Urban Land Surface Temperature: A Case Study of Dalian, China[J]. Urban Forestry & Urban Greening. 2017(22), 1–10.doi:10.1016/j.ufug.2017.01.002.

3.Spatiotemporal variations in greenspace ecosystem service value at urban fringes: A case study on Ganjingzi District in Dalian, China[J]. Science of the Total Environment. 639 (2018) 1453–1461,.doi:10.1016/j.scitotenv.2018.05.253.

(5) line 95-98, not needed, as you mention it in the end of this paper.

(6) for statistic analysis, the p-value, should not be capitalized.

(7) the conclusion section should be 5. Conclusion.

Reviewer #2: This paper explains the driving mechanism of the effects of urban green spaces on human health through structural equation models (SEMs). It has interesting findings such as that intensity (types of activities people engage in urban green spaces) matters the most to human health improvement. I have some comments and suggestions for the authors to consider to further improve the paper.

First, while this is an interesting paper examines an important question, I am not clear about the contribution the authors made to the field. The authors mentioned that they use Zhang et al’s model as the theoretical backbone. If so, please explain Zhang et al’s model in detail and point out what improvement the authors have made. Currently, it is not clearly if the authors’ model is any different from Zhange et al.

Second, the authors shall provide justification to their models and data variables. The biggest difference of this paper’s data variables, as the authors mentioned, is that they use people’s knowledge of UGS and effects on health, rather than other classical metrics. While this has certain advantages, such as simple and easy to measure, it also has subjective bias. The authors shall discuss this limitation. It also needs to provide justification why they prioritize people’s own perceptions/judgement over more objective measures. On another issue, the authors used education as a moderator, but I cannot find any reference or any justification in the text why education shall be considered as a moderator.

Finally, health response is measured subjectively and shall be renamed as “perceived health improvement” or other similar name.

Minor issues:

female-to-male proportion of 52% and 48%, it shall be 52:48 (2.2 study area)

6. PLOS authors have the option to publish the peer review history of their article (what does this mean?). If published, this will include your full peer review and any attached files.

Reviewer #1: No

Reviewer #2: No

---

## [Author Response · Author response to Decision Letter 0]

21 Jul 2020

Response letter to reviewers’ comments

PONE-D-20-17738

Testing a model explaining the mechanism driving the effects of urban green spaces on human health

PLOS ONE

Dear Prof. Jun Yang – Academic Editor,

Thank you for submitting your manuscript to PLOS ONE. After careful consideration, we feel that it has merit but does not fully meet PLOS ONE’s publication criteria as it currently stands. Therefore, we invite you to submit a revised version of the manuscript that addresses the points raised during the review process.

Authors: Thank you for allowing a revision of our manuscript. We have now revised thoroughly this manuscript following closely comments and suggestions from reviewers as detailed below:

Reviewer 1:

Overall, this paper is well-written and the topic is interesting. However, I suggest authors do some necessary revisions.

Authors: Thank you

(1) Overall, the English is good, while some expression is informal. On one hand, on the other....

Authors: We thank the reviewer for pointing this out. However, we maintain the expressions “on hand” and “on the other hand” since this was used only four times in the entire text. 

(2) The Title of this paper should be better if you use 'The mechanism driving the effects of urban green spaces on human health: based on structural equation models'

Authors: We have revised the title as follow: "Effects of Urban Green Spaces on Human Perceived Health Improvements: Provision of Green Spaces is not Enough but How People Use Them Matters".

(3) The abstract is actually not well written. line 12-19, authors focus on the model results, while it requires your efforts to translate into the general findings, as some readers are not familiar with the models.

Authors: In our revision, we make sure we did away with “models 1, 2, 3….etc.”; rather we insisted on SEM. Our initial statement reads as follows. "We show that our models 2 (where Provision is approximated as UGS quantity, and Exposure as duration), 3 (where Provision is approximated as quantity, and Exposure as intensity) and 12 (where Provision is approximated as distance of the closest UGS from people’s house, and Exposure as intensity) are equally the best of all 12 models tested. However, apart from model 2 that has no significant path, models 3 and 12 have the same significant path (motivation ~ intensity; β=7.86±2.03, P=0.0002), suggesting that visits to UGS may be motivated by opportunities of physical activities offered by UGS". 

This initial statement has been revised as follows: 

"We show that SEMs where i) Provision is approximated as UGS quantity, and Exposure as duration (SEMi), ii) Provision is approximated as quantity, and Exposure as intensity (SEMii) and iii) Provision is approximated as distance of the closest UGS from people’s house, and Exposure as intensity (SEMiii) are equally the best of all 12 SEMs tested. However, apart from the SEMi that has no significant path, SEMii and SEMiii have the same significant path (motivation ~ intensity; β=7.86±2.03, P=0.0002), suggesting that visits to UGS may be motivated by opportunities of physical activities offered by UGS. In all our scenarios, the best SEM is always the one where Exposure is measured as intensity, irrespective of how Provision is approximated"

(4) Line 30-35, the description is too short, you may try to describe in detail, especially the human health effect (e.g. noise reduction, urban heat island mitigation, urban flooding management, air and water purification). This is because your paper is focusing on the health effect.

You may refer:

1. The maintenance of prefabricated green roofs for preserving cooling performance: A field measurement in the subtropical city of Hangzhou, China. Sustainable Cities and Society, 61, 102314.

2. Assessing the Impacts of Urbanization-Associated Green Space on Urban Land Surface Temperature: A Case Study of Dalian, China[J]. Urban Forestry & Urban Greening. 2017(22), 1–10.doi:10.1016/j.ufug.2017.01.002.

3.Spatiotemporal variations in greenspace ecosystem service value at urban fringes: A case study on Ganjingzi District in Dalian, China[J]. Science of the Total Environment. 639 (2018) 1453–1461,.doi:10.1016/j.scitotenv.2018.05.253.

Authors: We have furthered our background information on UGS using the literature suggested by the reviewer. See first paragraph of the Introduction in the revised manuscript.

(5) line 95-98, not needed, as you mention it in the end of this paper.

Authors: We maintain the Ethics declaration here in the Method as this is a requirement of the Journal; then we deleted it from the end of the paper.

(6) for statistic analysis, the p-value, should not be capitalize

Authors: We have amended this as suggested. Thank you.

Reviewer 2

This paper explains the driving mechanism of the effects of urban green spaces on human health through structural equation models (SEMs). It has interesting findings such as that intensity (types of activities people engage in urban green spaces) matters the most to human health improvement. 

Authors: Thank you.

I have some comments and suggestions for the authors to consider to further improve the paper.

First, while this is an interesting paper examines an important question, I am not clear about the contribution the authors made to the field. The authors mentioned that they use Zhang et al’s model as the theoretical backbone. If so, please explain Zhang et al’s model in detail and point out what improvement the authors have made. Currently, it is not clearly if the authors’ model is any different from Zhange et al.

Authors: Indeed, we use Zhang et al.’s model as the backbone for the study. The reviewer asked for the explanation of Zhang et al.’s model and how our study improves the model. Zhang et al.’s model is explained in the paragraph from Lines 44 to 64 and the relationships among variables in the model are illustrated in Figure 1. In term of contribution of our study to the field, the different points of gaps in the model and thus the points of contribution of our study are presented in details in the Introduction section from lines 66 to 87. Theses points are discussed in the section 4.1 of the Discussion. For example, Zhang et al.’s model was never tested empirically. Our study fills that gap. Also, in Zhang et al.’s model there was no path linking directly UGS provision to health response. We tested this and confirm there was no need for it. Zhang et al.’s model was based on published studies that used quantitative data collected using technologies; here we collected data based on people’s perceptions and assessment of their health conditions. These key points of contribution of our study to the field and these points are highlighted in the Discussion section.

Second, the authors shall provide justification to their models and data variables. The biggest difference of this paper’s data variables, as the authors mentioned, is that they use people’s knowledge of UGS and effects on health, rather than other classical metrics. While this has certain advantages, such as simple and easy to measure, it also has subjective bias. The authors shall discuss this limitation. It also needs to provide justification why they prioritize people’s own perceptions/judgement over more objective measures. 

Authors We prioritize people’s perception for the following simple reasons: 

• In tropical and poor countries where lab. equipment and financial support for research are limited but the need for healthy cities is growing, using a questionnaire-based data collection can be easily used to investigate UGS-health relationships and promote the establishment of UGS in urban areas of those poor countries;

• With technology, we can have the most accurate data on green spaces in term of provision (numbers) and even amount of vegetation or green (using NDVI for example) in a given locality. However, if local people do not know about the existence of these green spaces in the area, if people are not even aware of potential effects of the green spaces on their health, they are not going to use them. That is why people’s own knowledge and perceptions of green spaces become important for them to use the green spaces and eventually benefit the positive effects of the green spaces on their health. The bottom line is that people cannot use something they don’t know they have in their area and they cannot use something they know nothing about (in term of benefit). These details are provided in the section “2.3.3. Mean of data collection and justification of the approach used”.

• We have also acknowledged the limitation of our approach as follows: "Nonetheless, we acknowledge that, due to the way health response was measured in the present study, our health response may not necessarily correspond always to the diagnostic on a medical report, hence we referred to it as perceived health conditions by the respondents".

On another issue, the authors used education as a moderator, but I cannot find any reference or any justification in the text why education shall be considered as a moderator.

Authors: Reference was provided for the choice of education as moderator in the section “2.3.2. Data collected”. This reference is Zhang et al. (2017). Moderators are factors that alter the strength of the relationships between different variables in the model. Zhang et al. (2017) suggested that socio-demographic factors may play the role of moderator, e.g. education level. In the case of education, this is because education level may be determinant of people’s level of understanding of UGS values, and this understanding would determine how consistent they are in using or not UGS. This information is now provided in the revised manuscript.

Finally, health response is measured subjectively and shall be renamed as “perceived health improvement” or other similar name.

Authors: We fully agree with the reviewer and we have now specified “perceived health” very clearly in the title.

Minor issues:

female-to-male proportion of 52% and 48%, it shall be 52:48 (2.2 study area)

 Authors: This has been corrected.

---

## [Decision Letter · Decision Letter 1]

4 Sep 2020

Effects of Urban Green Spaces on Human Perceived Health Improvements: Provision of Green Spaces is not Enough but How People Use Them Matters

PONE-D-20-17738R1

Dear Dr. Yessoufou,

We’re pleased to inform you that your manuscript has been judged scientifically suitable for publication and will be formally accepted for publication once it meets all outstanding technical requirements.

Kind regards,

Jun Yang

Academic Editor

PLOS ONE

Additional Editor Comments (optional):

Accept

Reviewer 1：

The author has revised all the suggestions. I recommend publishing after minor revison.

Please correct Ref. 7. These authors should be Yang, J., sun, J., Ge, Q., Li, X.

Reviewer 3

Well done, authors have addressed all my questions. The title is more proper and interesting than I suggested.

Reviewers' comments:

Reviewer's Responses to Questions

**Comments to the Author**

1. If the authors have adequately addressed your comments raised in a previous round of review and you feel that this manuscript is now acceptable for publication, you may indicate that here to bypass the “Comments to the Author” section, enter your conflict of interest statement in the “Confidential to Editor” section, and submit your "Accept" recommendation.

Reviewer #1: All comments have been addressed

Reviewer #3: All comments have been addressed

2. Is the manuscript technically sound, and do the data support the conclusions?

Reviewer #1: Yes

Reviewer #3: Yes

3. Has the statistical analysis been performed appropriately and rigorously? 

Reviewer #1: Yes

Reviewer #3: Yes

4. Have the authors made all data underlying the findings in their manuscript fully available?

Reviewer #1: Yes

Reviewer #3: Yes

5. Is the manuscript presented in an intelligible fashion and written in standard English?

Reviewer #1: Yes

Reviewer #3: Yes

6. Review Comments to the Author

Reviewer #1: Well done, authors have addressed all my questions. The title is more proper and interesting than I suggested.

Reviewer #3: The author has revised all the suggestions. I recommend publishing after minor revison.

Please correct Ref. 7. These authors should be Yang, J., sun, J., Ge, Q., Li, X.

7. PLOS authors have the option to publish the peer review history of their article (what does this mean?). If published, this will include your full peer review and any attached files.

Reviewer #1: No

Reviewer #3: No

---

## [Editor Report · Acceptance letter]

14 Sep 2020

PONE-D-20-17738R1 

Effects of Urban Green Spaces on Human Perceived Health Improvements: Provision of Green Spaces is not Enough but How People Use Them Matters 

Dear Dr. Yessoufou:

I'm pleased to inform you that your manuscript has been deemed suitable for publication in PLOS ONE. Congratulations! Your manuscript is now with our production department. 

Kind regards, 

on behalf of

Dr. Jun Yang 

Academic Editor

PLOS ONE